# BIG & QUIC: Sparse Inverse Covariance Estimation for a Million Variables

**Cho-Jui Hsieh, Mátyás A. Sustik, Inderjit S. Dhillon, Pradeep Ravikumar**
Department of Computer Science
University of Texas at Austin
{cjhsieh,sustik,inderjit,pradeepr}@cs.utexas.edu

**Russell A. Poldrack**
Department of Psychology and Neurobiology
University of Texas at Austin
poldrack@mail.utexas.edu

## Abstract

The $\ell_1$-regularized Gaussian maximum likelihood estimator (MLE) has been shown to have strong statistical guarantees in recovering a sparse inverse covariance matrix even under high-dimensional settings. However, it requires solving a difficult non-smooth log-determinant program with number of parameters scaling quadratically with the number of Gaussian variables. State-of-the-art methods thus do not scale to problems with more than $20,000$ variables. In this paper, we develop an algorithm BIGQUIC, which can solve 1 million dimensional $\ell_1$-regularized Gaussian MLE problems (which would thus have 1000 billion parameters) using *a single machine*, with bounded memory. In order to do so, we carefully exploit the underlying structure of the problem. Our innovations include a novel block-coordinate descent method with the blocks chosen via a clustering scheme to minimize repeated computations; and allowing for *inexact* computation of specific components. In spite of these modifications, we are able to theoretically analyze our procedure and show that BIGQUIC can achieve super-linear or even quadratic convergence rates.

## 1  Introduction

Let $\{\mathbf{y}_1, \mathbf{y}_2, \ldots, \mathbf{y}_n\}$ be $n$ samples drawn from a $p$-dimensional Gaussian distribution $\mathcal{N}(\mu, \Sigma)$, also known as a Gaussian Markov Random Field (GMRF). An important problem is that of recovering the covariance matrix (or its inverse) of this distribution, given the $n$ samples, in a high-dimensional regime where $n \ll p$. A popular approach involves leveraging the structure of sparsity in the inverse covariance matrix, and solving the following $\ell_1$-regularized maximum likelihood problem:

$$\arg\min_{\Theta \succ 0}\{-\log\det\Theta + \mathrm{tr}(S\Theta) + \lambda\|\Theta\|_1\} = \arg\min_{\Theta \succ 0} f(\Theta), \tag{1}$$

where $S = \frac{1}{n}\sum_{i=1}^{n}(\mathbf{y}_i - \tilde{\mu})(\mathbf{y}_i - \tilde{\mu})^T$ is the sample covariance matrix and $\tilde{\mu} = \frac{1}{n}\sum_{i=1}^{n}\mathbf{y}_i$ is the sample mean. While the non-smooth log-determinant program in (1) is usually considered a difficult optimization problem to solve, due in part to its importance, there has been a long line of recent work on algorithms to solve (1): see [7, 6, 3, 16, 17, 18, 15, 11] and references therein. The state-of-the-art seems to be a second order method QUIC [9] that has been shown to achieve super-linear convergence rates. Complementary techniques such as exact covariance thresholding [13, 19], and the divide and conquer approach of [8], have also been proposed to speed up the solvers. However, as noted in [8], the above methods do not scale to problems with more than $20,000$ variables, and typically require several hours even for smaller dimensional problems involving ten thousand variables. There has been some interest in statistical estimators other than (1) that are more amenable to optimization: including solving node-wise Lasso regression problems [14] and the separable linear program based CLIME estimator [2]. However the caveat with these estimators is that they are not guaranteed to yield a positive-definite covariance matrix, and typically yield less accurate parameters.

What if we want to solve the $M$-estimator in (1) with a million variables? Note that the number of parameters in (1) is quadratic in the number of variables, so that for a million variables, we would

have a trillion parameters. There has been considerable recent interest in such "Big Data" problems involving large-scale optimization: these however are either targeted to "big-n" problems with a lot of samples, unlike the constraint of "big-p" with a large number of variables in our problem, or are based on large-scale distributed and parallel frameworks, which require a cluster of processors, as well as software infrastructure to run the programs over such clusters. At least one caveat with such large-scale distributed frameworks is that they would be less amenable to exploratory data analysis by "lay users" of such GMRFs. Here we ask the following ambitious but simple question: can we solve the $M$-estimator in (1) with a million variables using a single machine with bounded memory? This might not seem like a viable task at all in general, but note that the optimization problem in (1) arises from a very structured statistical estimation problem: can we leverage the underlying structure to be able to solve such an ultra-large-scale problem?

In this paper, we propose a new solver, BIGQUIC, to solve the $\ell_1$-regularized Gaussian MLE problem with extremely high dimensional data. Our method can solve one million dimensional problems with 1000 billion variables using a single machine with 32 cores and 32G memory. Our proposed method is based on the state-of-the-art framework of QUIC [9, 8]. The key bottleneck with QUIC stems from the memory required to store the gradient $W = X^{-1}$ of the iterates $X$, which is a dense $p \times p$ matrix, and the computation of the log-determinant function of a $p \times p$ matrix. A starting point to reduce the memory footprint is to use sparse representations for the iterates $X$ and compute the elements of the empirical covariance matrix $S$ on demand from the sample data points. In addition we also have to avoid the storage of the dense matrix $X^{-1}$ and perform intermediate computations involving functions of such dense matrices on demand. These naive approaches to reduce the memory however would considerably increase the computational complexity, among other caveats, which would make the algorithm highly impractical.

To address this, we present three key innovations. Our first is to carry out the coordinate descent computations in a blockwise manner, and by selecting the blocks very carefully using an automated clustering scheme, we not only leverage sparsity of the iterates, but help cache computations suitably. Secondly, we reduce the computation of the log-determinant function to linear equation solving using the Schur decomposition that also exploits the symmetry of the matrices in question. Lastly, since the Hessian computation is a key bottleneck in the second-order method, we compute it *inexactly*. We show that even with these modifications and inexact computations, we can still guarantee not only convergence of our overall procedure, but can easily control the degree of approximation of Hessian to achieve super-linear or even quadratic convergence rates. Inspite of our low-memory footprint, these innovations allow us to beat the state of the art DC-QUIC algorithm (which has no memory limits) in computational complexity even on medium-size problems of a few thousand variables. Finally, we show how to parallelize our method in a multicore shared memory system.

The paper is organized as follows. In Section 2, we briefly review the QUIC algorithm and outline the difficulties of scaling QUIC to million dimensional data. Our algorithm is proposed in Section 3. We theoretically analyze our algorithm in Section 4, and present experimental results in Section 5.

## 2   Difficulties in scaling QUIC to million dimensional data

Our proposed algorithm is based on the framework of QUIC [9]; which is a state of the art procedure for solving (1), based on a second-order optimization method. We present a brief review of the algorithm, and then explain the key bottlenecks that arise when scaling it to million dimensions. Since the objective function of (1) is non-smooth, we can separate the smooth and non-smooth part by $f(X) = g(X) + h(X)$, where $g(X) = -\log \det X + \operatorname{tr}(SX)$ and $h(X) = \lambda \|X\|_1$.

QUIC is a second-order method that iteratively solves for a generalized Newton direction using coordinate descent; and then descends using this generalized Newton direction and line-search. To leverage the sparsity of the solution, the variables are partitioned into $S_{fixed}$ and $S_{free}$ sets:

$$X_{ij} \in S_{fixed} \text{ if } |\nabla_{ij}g(X)| \leq \lambda_{ij}, \text{ and } X_{ij} = 0, \quad X_{ij} \in S_{free} \text{ otherwise.} \quad (2)$$

Only the *free* set $S_{free}$ is updated at each Newton iteration, reducing the number of variables to be updated to $m = |S_{free}|$, which is comparable to $\|X^*\|_0$, the sparsity of the solution.

**Difficulty in Approximating the Newton Direction.**   Let us first consider the generalized Newton direction for (1):

$$D_t = \arg \min_D \{\bar{g}_{X_t}(D) + h(X_t + D)\}, \quad (3)$$

where

$$\bar{g}_{X_t}(D) = g(X_t) + \operatorname{tr}(\nabla g(X_t)^T D) + \frac{1}{2} \operatorname{vec}(D)^T \nabla^2 g(X_t) \operatorname{vec}(D). \quad (4)$$

In our problem $\nabla g(X_t) = S - X_t^{-1}$ and $\nabla^2 g(X) = X_t^{-1} \otimes X_t^{-1}$, where $\otimes$ denotes the Kronecker product of two matrices. When $X_t$ is sparse, the Newton direction computation (3) can be solved

efficiently by coordinate descent [9]. The obvious implementation calls for the computation and storage of $W_t = X_t^{-1}$; using this to compute $a = W_{ij}^2 + W_{ii}W_{jj}$, $b = S_{ij} - W_{ij} + \mathbf{w}_i^T D \mathbf{w}_j$, and $c = X_{ij} + D_{ij}$. Armed with these quantities, the coordinate descent update for variable $D_{ij}$ takes the form:

$$D_{ij} \leftarrow D_{ij} - c + \mathcal{S}(c - b/a, \lambda_{ij}/a), \tag{5}$$

where $\mathcal{S}(z, r) = \text{sign}(z) \max\{|z| - r, 0\}$ is the soft-thresholding function.

The **key computational bottleneck** here is in computing the terms $\mathbf{w}_i^T D \mathbf{w}_j$, which take $O(p^2)$ time when implemented naively. To address this, [9] proposed to store and maintain $U = DW$, which reduced the cost to $O(p)$ flops per update. However, this is not a strategy we can use when dealing with very large data sets: storing the $p$ by $p$ dense matrices $U$ and $W$ in memory would be prohibitive. The straightforward approach is to compute (and recompute when necessary) the elements of $W$ on demand, resulting in $O(p^2)$ time complexity.

Our key innovation to address this is a novel block coordinate descent scheme, detailed in Section 3.1, that also uses clustering to strike a balance between memory use and computational cost while exploiting sparsity. The result is a procedure with comparable wall-time to that of QUIC on mid-sized problems and can scale up to very large problem instances that the original QUIC could not.

**Difficulty in the Line Search Procedure.** After finding the generalized Newton direction $D_t$, QUIC then descends using this direction after a line-search via Armijo's rule. Specifically, it selects the largest step size $\alpha \in \{\beta^0, \beta^1, \dots\}$ such that $X + \alpha D_t$ is (a) positive definite, and (b) satisfies the following sufficient decrease condition:

$$f(X + \alpha D^*) \leq f(X) + \alpha \sigma \delta, \ \ \delta = \text{tr}(\nabla g(X)^T D^*) + \|X + D^*\|_1 - \|X\|_1. \tag{6}$$

The **key computational bottleneck** is checking positive definiteness (typically by computing the smallest eigenvalue), and the computation of the determinant of a sparse matrix with dimension that can reach a million. As we show in Appendix 6.4, the time and space complexity of classical sparse Cholesky decomposition generally grows quadratically to dimensionality even when fixing the number of nonzero elements in the matrix, so it is nontrivial to address this problem. Our key innovation, detailed in Section 3.2, is an efficient procedure that checks both conditions (a) and (b) above using Schur complements and sparse linear equation solving. The computation only uses memory proportional to the number of nonzeros in the iterate.

Many other difficulties arise when dealing with large sparse matrices in the sparse inverse covairance problem. We present some of them in Appendix 6.5.

## 3 Our proposed algorithm

In this section, we describe our proposed algorithm, BIGQUIC, with the key innovations mentioned in the previous section. We assume that the iterates $X_t$ have $m$ nonzero elements, and that each iterate is stored in memory using a sparse format. We denote the size of the free set by $s$ and observe that it is usually very small and just a constant factor larger than $m^*$, the number of nonzeros in the final solution [9]. Also, the sample covariance matrix is stored in its factor form $S = YY^T$, where $Y$ is the normalized sample matrix. We now discuss a crucial element of BIGQUIC, our novel block coordinate descent scheme for solving each subproblem (3).

### 3.1 Block Coordinate Descent method

The most expensive step during the coordinate descent update for $D_{ij}$ is the computation of $\mathbf{w}_i^T D \mathbf{w}_j$, where $w_i$ is the $i$-th column of $W = X^{-1}$; see (5). It is not possible to compute $W = X^{-1}$ with Cholesky factorization as was done in [9], nor can it be stored in memory. Note that $\mathbf{w}_i$ is the solution of the linear system $X \mathbf{w}_i = \mathbf{e}_i$. We thus use the conjugate gradient method (CG) to compute $\mathbf{w}_i$, leveraging the fact that $X$ is a positive definite matrix. This solver requires only matrix vector products, which can be efficiently implemented for the sparse matrix $X$. CG has time complexity $O(mT)$, where $T$ is the number of iterations required to achieve the desired accuracy.

**Vanilla Coordinate Descent.** A single step of coordinate descent requires the solution of two linear systems $X \mathbf{w}_i = \mathbf{e}_i$ and $X \mathbf{w}_j = \mathbf{e}_j$ which yield the vectors $\mathbf{w}_i, \mathbf{w}_j$, and we can then compute $\mathbf{w}_i^T D \mathbf{w}_j$. The time complexity for each update would require $O(mT + s)$ operations, and the overall complexity will be $O(msT + s^2)$ for one full sweep through the entire matrix. Even when the matrix is sparse, the quadratic dependence on nonzero elements is expensive.

**Our Approach: Block Coordinate Descent with memory cache scheme.** In the following we present a block coordinate descent scheme that can accelerate the update procedure by storing and

reusing more results of the intermediate computations. The resulting increased memory use and speedup is controlled by the number of blocks employed, that we denote by $k$.

Assume that only some columns of $W$ are stored in memory. In order to update $D_{ij}$, we need both $\mathbf{w}_i$ and $\mathbf{w}_j$; if either one is not directly available, we have to recompute it by CG and we call this a "cache miss". A good update sequence can minimize the cache miss rate. While it is hard to find the optimal sequence in general, we successfully applied a *block by block* update sequence with a careful clustering scheme, where the number of cache misses is sufficiently small.

Assume we pick $k$ such that we can store $p/k$ columns of $W$ ($p^2/k$ elements) in memory. Suppose we are given a partition of $\mathcal{N} = \{1, \ldots, p\}$ into $k$ blocks, $S_1, \ldots, S_k$. We divide matrix $D$ into $k \times k$ blocks accordingly. Within each block we run $T_{\text{inner}}$ sweeps over variables within that block, and in the outer iteration we sweep through all the blocks $T_{\text{outer}}$ times. We use the notation $W_{S_q}$ to denote a $p$ by $|S_q|$ matrix containing columns of $W$ that corresponds to the subset $S_q$.

**Coordinate descent within a block.** To update the variables in the block $(S_z, S_q)$ of $D$, we first compute $W_{S_z}$ and $W_{S_q}$ by CG and store it in memory, meaning that there is no cache miss during the within-block coordinate updates. With $U_{s_q} = DW_{S_q}$ maintained, the update for $D_{ij}$ can be computed by $\mathbf{w}_i^T \mathbf{u}_j$ when $i \in S_z$ and $j \in S_q$. After updating each $D_{ij}$ to $D_{ij} + \mu$, we can maintain $U_{S_q}$ by

$$U_{it} \leftarrow U_{it} + \mu W_{jt}, \ \ U_{jt} \leftarrow U_{jt} + \mu W_{it}, \ \ \forall t \in S_q.$$

The above coordinate update computations cost only $O(p/k)$ operations because we only update a subset of the columns. Observe that $U_{rt}$ never changes when $r \notin \{S_z \cup S_q\}$.

Therefore, we can use the following arrangement to further reduce the time complexity. Before running coordinate descent for the block we compute and store $P_{ij} = (\mathbf{w}_i)_{S_{\bar{z}\bar{q}}}^T (\mathbf{u}_j)_{S_{\bar{z}\bar{q}}}$ for all $(i, j)$ in the free set of the current block, where $S_{\bar{z}\bar{q}} = \{i \mid i \notin S_z \text{ and } i \notin S_q\}$. The term $\mathbf{w}_i^T \mathbf{u}_j$ for updating $D_{ij}$ can then be computed by $\mathbf{w}_i^T \mathbf{u}_j = P_{ij} + \mathbf{w}_{S_z}^T \mathbf{u}_{S_z} + \mathbf{w}_{S_q}^T \mathbf{u}_{S_q}$. With this trick, each coordinate descent step within the block only takes $O(p/k)$ time, and we only need to store $U_{S_z, S_q}$, which only requires $O(p^2/k^2)$ memory. Computing $P_{ij}$ takes $O(p)$ time for each $i, j$, so if we update each coordinate $T_{\text{inner}}$ times within a block, the time complexity is $O(p + T_{\text{inner}}p/k)$ and the amortized cost per coordinate update is only $O(p/T_{\text{inner}} + p/k)$. This time complexity suggests that we should run more iterations within each block.

**Sweeping through all the blocks.** To go through all the blocks, each time we select a $z \in \{1, \ldots, k\}$ and updates blocks $(S_z, S_1), \ldots, (S_z, S_k)$. Since all of them share $\{\mathbf{w}_i \mid i \in S_z\}$, we first compute them and store in memory. When updating an off-diagonal block $(S_z, S_q)$, if the free sets are dense, we need to compute and store $\{\mathbf{w}_i \mid i \in S_q\}$. So totally each block of $W$ will be computed $k$ times. The total time complexity becomes $O(kpmT)$, where $m$ is number of nonzeros in $X$ and $T$ is number of conjugate gradient iterations. Assume the nonzeros in $X$ is close to the size of free set ($m \approx s$), then each coordinate update costs $O(kpT)$ flops.

**Selecting the blocks using clustering.** We now show that a careful selection of the blocks using a clustering scheme can lead to dramatic speedup for block coordinate descent. When updating variables in the block $(S_z, S_q)$, we would need the column $\mathbf{w}_j$ only if some variable in $\{D_{ij} \mid i \in S_z\}$ lies in the free set. Leveraging this key observation, given two partitions $S_z$ and $S_q$, we define the set of *boundary nodes* as: $B(S_z, S_q) \equiv \{j \mid j \in S_q \text{ and } \exists i \in S_z \text{ s.t. } F_{ij} = 1\}$, where the matrix $F$ is an indicator of the free set.

The number of columns to be computed in one sweep is then given by $p + \sum_{z \neq q} |B(S_z, S_q)|$. Therefore, we would like to find a partition $\{S_1, \ldots, S_k\}$ for which $\sum_{z \neq q} |B(S_z, S_q)|$ is minimal. It appears to be hard to find the partitioning that minimizes the number of boundary nodes. However, we note that the number in question is bounded by the number of cross cluster edges: $B(S_z, S_q) < \sum_{i \in S_z, j \in S_q} F_{ij}$. This suggests the use of graph clustering algorithms, such as METIS [10] or Graclus [5] which minimize the right hand side. Assuming that the ratio of between-cluster edges to the number of total edges is $r$, we observe a reduced time complexity of $O((p+rm)T)$ when computing elements of $W$, and $r$ is very small in real datasets. In real datasets, when we converge to very sparse solutions, more than 95% of edges are in the diagonal blocks. In case of the fMRI dataset with $p = 228483$, we used 20 blocks, and the total number of boundary nodes were only $|B| = 8697$. Compared to block coordinate descent with random partition, which generally needs to compute $228483 \times 20$ columns, the clustering resulted in the computation of $228483 + 8697$ columns, thus achieved an almost 20 times speedup. In Appendix 6.6 we also discuss additional benefits of the graph clustering algorithm that results in accelerated convergence.

## 3.2 Line Search

The line search step requires an efficient and scalable procedure that computes $\log \det(A)$ and checks the positive definiteness of a sparse matrix $A$. We present a procedure that has complexity of at most $O(mpT)$ where $T$ is the number of iterations used by the sparse linear solver. We note that computing $\log \det(A)$ for a large sparse matrix $A$ for which we only have a matrix-vector multiplication subroutine available is an interesting subproblem on its own and we expect that numerous other applications may benefit from the approach presented below. The following lemma can be proved by induction on $p$:

**Lemma 1.** *If $A = \begin{pmatrix} a & \mathbf{b}^T \\ \mathbf{b} & C, \end{pmatrix}$ is a partitioning of an arbitrary $p \times p$ matrix, where $a$ is a scalar and $\mathbf{b}$ is a $p-1$ dimensional vector then $\det(A) = \det(C)(a - \mathbf{b}^T C^{-1} \mathbf{b})$. Moreover, $A$ is positive definite if and only if $C$ is positive definite and $(a - \mathbf{b}^T C^{-1} \mathbf{b}) > 0$.*

The above lemma allows us to compute the determinant by reducing it to solving linear systems; and also allows us to check positive-definiteness. Applying Lemma 1 recursively, we get

$$\log \det A = \sum_{i=1}^{p} \log(A_{ii} - A_{(i+1):p,i}^T A_{(i+1):p,(i+1):p}^{-1} A_{(i+1):p,i}), \tag{7}$$

where each $A_{i_1:i_2,j_1:j_2}$ denotes a submatrix of $A$ with row indexes $i_1, \ldots, i_2$ and column indexes $j_1, \ldots, j_2$. Each $A_{(i+1):p,(i+1):p}^{-1} A_{(i+1):p,i}$ in the above formula can be computed as the solution of a linear system and hence we can avoid the storage of the (dense) inverse matrix. By Lemma 1, we can check the positive definiteness by verifying that all the terms in (7) are positive definite. Notice that we have to compute (7) in a reverse order ($i = p, \ldots, 1$) to avoid the case that $A_{(i+1):p,(i+1):p}$ is non positive definite.

## 3.3 Summary of the algorithm

In this section we present BIGQUIC as Algorithm 1. The detailed time complexity analysis are presented in Appendix 6.7. In summary, the time needed to compute the columns of $W$ in block coordinate descent, $O((p + |B|)mTT_{\text{outer}})$, dominates the time complexity, which underscores the importance of minimizing the number of boundary nodes $|B|$ via our clustering scheme.

---

**Algorithm 1:** BIGQUIC algorithm

**Input** : Samples $Y$, regularization parameter $\lambda$, initial iterate $X_0$
**Output**: Sequence $\{X_t\}$ that converges to $X^*$.
**1 for** $t = 0, 1, \ldots$ **do**
**2**      Compute $W_t = X_t^{-1}$ column by column, partition the variables into free and fixed sets.
**3**      Run graph clustering algorithm based on absolute values on free set.
**4**      **for** *sweep* $= 1, \ldots, T_{outer}$ **do**
**5**          **for** $s = 1, \ldots, k$ **do**
**6**              Compute $W_{S_s}$ by CG.
**7**              **for** $q = 1, \ldots, k$ **do**
**8**                  Identify boundary nodes $B_{sq} := B(S_s, S_q) \subset S_q$ (only need if $s \neq q$)
**9**                  Compute $W_{B_{sq}}$ for boundary nodes (only need if $s \neq q$).
**10**                  Compute $U_{B_{sq}}$, and $P_{ij}$ for all $(i, j)$ the current block.
**11**                  Conduct coordinate updates.

**12**      Find the step size $\alpha$ by the method proposed in Section 3.2.

---

**Parallelization.** While our method can run well on a single machine with a single core, here we point out components of our algorithm that can be "embarrassingly" parallelized on any single machine with multiple cores (with shared memory). We first note that we can obtain a good starting point for our algorithm by applying the divide-and-conquer framework proposed in [8]: this divides the problem into $k$ subproblems, which can then be independently solved in parallel. Consider the steps of our Algorithm 1 in BIGQUIC. In step 2, instead of computing columns of $W$ one by one, we can compute $t$ rows of $W$ at a time, and parallelize these $t$ jobs. A similar trick can be used in step 6 and 9. In step 3, we use the multi-core version of METIS (ParMETIS) for graph clustering.

In step 8 and 10, the computations are naturally independent. In step 15, we compute each term in (7) independently and abort if any of the processes report non-positive definiteness. The only sequential part is the coordinate update in step 11, but note, (see Section 3.1), that we have reduced the complexity of this step from $O(p)$ in QUIC to $O(p/k)$.

## 4 Convergence Analysis

In this section, we present two main theoretical results. First, we show that our algorithm converges to the global optimum even with inaccurate Hessian computation. Second, we show that by a careful control of the error in the Hessian computation, BIGQUIC can still achieve a quadratic rate of convergence in terms of Newton iterations. Our analysis differs from that in QUIC [9], where the computations are all assumed to be accurate. [11] also provides a convergence analysis for general proximal Newton methods, but our algorithm with modifications such as fixed/free set selection does not exactly fall into their framework; moreover our analysis shows a quadratic convergence rate, while they only show a super-linear convergence rate.

In the BIGQUIC algorithm, we compute $\mathbf{w}_i$ in two places. The first place is the gradient computation in the second term of (4), where $\nabla g(X) = S - W$. The second place is in the third term of (4), where $\nabla^2 g(X) = W \otimes W$. At the first glance they are equivalent and can be computed simultaneously, but it turns out that by carefully analysing the difference between two types of $\mathbf{w}_i$, we can achieve much faster convergence, as discussed below.

The key observation is that we only require the gradient $W_{ij}$ for all $(i, j) \in S_{free}$ to conduct coordinate descent updates. Since the free set is very sparse and can fit in memory, those $W_{ij}$ only need to be *computed once* and stored in memory. On the other hand, the computation of $\mathbf{w}_i^T D \mathbf{w}_j$ corresponds to the Hessian computation, and we need two columns for each coordinate update, which has to be *computed repeatedly*.

It is easy to produce an example where the algorithm converges to a wrong point when the gradient computation is not accurate, as shown in Figure 5(b) (in Appendix 6.5). Luckily, based on the above analysis the gradient only needs to be computed once per Newton iteration, so we can compute it with high precision. On the other hand, $\mathbf{w}_i$ for the Hessian has to be computed repeatedly, so we do not want to spend too much time to compute each of them accurately. We define $\hat{H}_t = \hat{W}_t \otimes \hat{W}_t$ to be the approximated Hessian matrix, and derive the following theorem to show that even if Hessian is inaccurate, BIGQUIC still converges to the global optimum. Notice that our proof covers BIGQUIC algorithm with fixed/free set selection, and the only assumption is that subproblem (3) is solved exactly for each Newton iteration; it is the future work to consider the case where subproblems are solved approximately.

**Theorem 1.** *For solving* (1)*, if* $\nabla g(X)$ *is computed exactly and* $\bar{\eta} I \succeq \hat{H}_t \succeq \eta I$ *for some constant* $\bar{\eta}, \eta > 0$ *at every Newton iteration, then* BIGQUIC *converges to the global optimum.*

The proof is in Appendix 6.1. Theorem 1 suggests that we do not need very accurate Hessian computation for convergence. To have super-linear convergence rate, we require the Hessian computation to be more and more accurate as $X_t$ approaches $X^*$. We first introduce the following notion of minimum norm subgradient to measure the optimality of $X$:

$$\text{grad}_{ij}^S f(X) = \begin{cases} \nabla_{ij} g(X) + \text{sign}(X_{ij}) \lambda_{ij} & \text{if } X_{ij} \neq 0, \\ \text{sign}(\nabla_{ij} g(X)) \max(|\nabla_{ij} g(X)| - \lambda_{ij}, 0) & \text{if } X_{ij} = 0. \end{cases}$$

The following theorem then shows that if we compute Hessian more and more accurately, BIGQUIC will have a super-linear or even quadratic convergence rate.

**Theorem 2.** *When applying* BIGQUIC *to solve* (1)*, assume* $\nabla g(X_t)$ *is exactly computed and* $\nabla^2 g(X_t)$ *is approximated by* $H_t$*, and the following condition holds:*

$$\nexists(i, j) \text{ such that } X_{ij}^* = 0 \text{ and } |\nabla_{ij} g(X^*)| = \lambda. \tag{8}$$

*Then* $\|X_{t+1} - X^*\| = O(\|X_t - X^*\|^{1+p})$ *as* $k \to \infty$ *for* $0 < p \leq 1$ *if and only if*

$$\|\hat{H}_t - \nabla^2 g(X_t)\| = O(\|\text{grad}^S(X_t)\|^p) \text{ as } k \to \infty. \tag{9}$$

The proof is in Appendix 6.2. The assumption in (8) can be shown to be satisfied with very high probability (and was also satisfied in our experiments). Theorem 2 suggests that we can achieve super-linear, or even quadratic convergence rate by a careful control of the approximated Hessian $\hat{H}_t$. In the BIGQUIC algorithm, we can further control $\|\hat{H}_t - \nabla^2 g(X_t)\|$ by the residual of conjugate

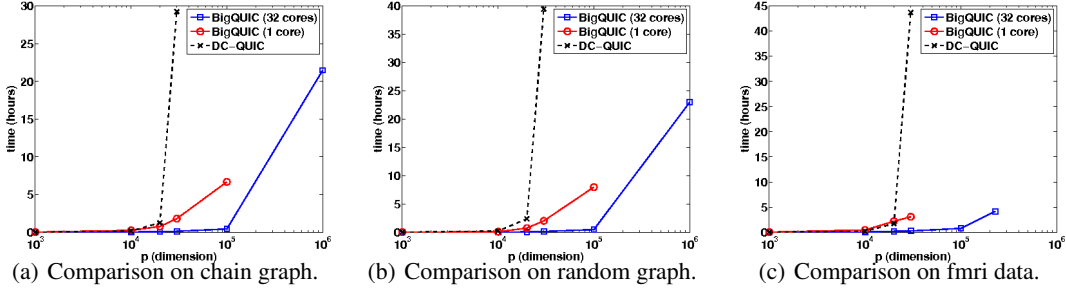

(a) Comparison on chain graph.  (b) Comparison on random graph.  (c) Comparison on fmri data.

Figure 1: The comparison of scalability on three types of graph structures. In all the experiments, BIGQUIC can solve larger problems than QUIC even with a single core, and using 32 cores BIGQUIC can solve million dimensional data in one day.

gradient solvers to achieve desired convergence rate. Suppose the residual is $b_i = X\hat{\mathbf{w}}_i - \mathbf{e}_i$ for each $i = 1, \ldots, p$, and $B_t = [b_1 b_2 \ldots b_p]$ is a collection of the residuals at the $t$-th iteration. The following theorem shows that we can control the convergence rate by controlling the norm of $B_t$.

**Theorem 3.** *In the* BIGQUIC *algorithm, if the residual matrix* $\|B_t\| = O(\|\operatorname{grad}^S(X_t)\|^p)$ *for some* $0 < p \leq 1$ *as* $t \to \infty$*, then* $\|X_{t+1} - X^*\| = O(\|X_t - X^*\|^{1+p})$ *as* $t \to \infty$.

The proof is in Appendix 6.3. Since $\operatorname{grad}^S(X_t)$ can be easily computed without additional cost, and residuals $B$ can be naturally controlled when running conjugate gradient, we can easily control the asymptotic convergence rate in practice.

## 5  Experimental Results

In this section, we show that our proposed method BIGQUIC can scale to high-dimensional datasets on both synthetic data and real data. All the experiments are run on a single computing node with 4 Intel Xeon E5-4650 2.7GHz CPUs, each with 8 cores and 32G memory.

**Scalability of** BIGQUIC **on high-dimensional datasets.**  In the first set of experiments, we show BIGQUIC can scale to extremely high dimensional datasets. We conduct experiments on the following synthetic and real datasets:
(1) Chain graphs: the ground truth precision matrix is set to be $\Sigma^{-1}_{i,i-1} = -0.5$ and $\Sigma^{-1}_{i,i} = 1.25$.
(2) Graphs with random pattern: we use the procedure mentioned in Example 1 in [12] to generate random pattern. When generating the graph, we assume there are 500 clusters, and 90% of the edges are within clusters. We fix the average degree to be 10.
(3) FMRI data: The original dataset has dimensionality $p = 228,483$ and $n = 518$. For scalability experiments, we subsample various number of random variables from the whole dataset.

We use $\lambda = 0.5$ for chain and random Graph so that number of recovered edges is close to the ground truth, and set number of samples $n = 100$. We use $\lambda = 0.6$ for the fMRI dataset, which recovers a graph with average degree 20. We set the stopping condition to be $\operatorname{grad}^S(X_t) < 0.01\|X_t\|_1$. In all of our experiments, number of nonzeros during the optimization phase do not exceed $5\|X^*\|_0$ in intermediate steps, therefore we can always store the sparse representation of $X_t$ in memory. For BIGQUIC, we set blocks $k$ to be the smallest number such that $p/k$ columns of $W$ can fit into 32G memory. For both QUIC and BIGQUIC, we apply the divide and conquer method proposed in [8] with 10-clusters to get a better initial point. The results are shown in Figure 1. We can see that BIGQUIC can solve one million dimensional chain graphs and random graphs in one day, and handle the full fMRI dataset in about 5 hours.

More interestingly, even for dataset with size less than 30000, where $p^2$ size matrices can fit in memory, BIGQUIC is faster than QUIC by exploiting the sparsity. Figure 2 shows an example on a sampled fMRI dataset with $p = 20000$, and we can see BIGQUIC outperforms QUIC even when using a single core. Also, BIGQUIC is much faster than other solvers, including Glasso [7] and ALM [17]. Figure 3 shows the speedup under a multicore shared memory machine. BIGQUIC can achieve about 14 times speedup using 16 cores, and 20 times speedup when using 32 cores.

**FMRI dataset.**  An extensive resting state fMRI dataset from a single individual was analyzed in order to test BIGQUIC on real-world data. The data (collected as part of the MyConnectome project: `http://www.myconnectome.org`) comprised 36 resting fMRI sessions collected across different days using whole-brain multiband EPI acquisition, each lasting 10 minutes (TR=1.16 secs, multiband factor=4, TE = 30 ms, voxel size = 2.4 mm isotropic, 68 slices, 518 time points). The

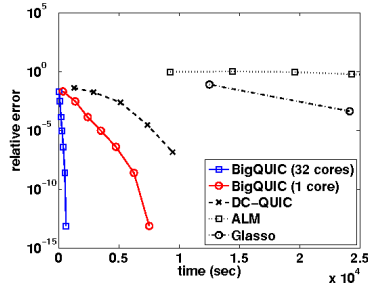

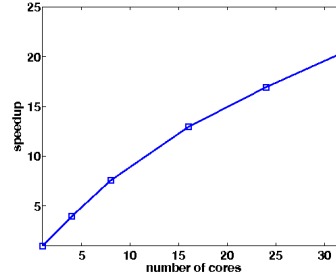

Figure 2: Comparison on fMRI data with $p = 20000$ (the maximum dimension that state-of-the-art softwares can handle).

Figure 3: The speedup of BIGQUIC when using multiple cores.

data were preprocessed using FSL 5.0.2, including motion correction, scrubbing of motion frames, registration of EPI images to a common high-resolution structural image using boundary-based registration, and affine transformation to MNI space. The full brain mask included 228,483 voxels. After motion scrubbing, the dataset included a total 18,435 time points across all sessions.

BIGQUIC was applied to the full dataset: for the first time, we can learn a GMRF over the entire set of voxels, instead of over a smaller set of curated regions or supervoxels. Exploratory analyses over a range of $\lambda$ values suggested that $\lambda = 0.5$ offered a reasonable level of sparsity. The resulting graph was analyzed to determine whether it identified neuroscientifically plausible networks. Degree was computed for each vertex; high-degree regions were primarily found in gray matter regions, suggesting that the method successfully identified plausible functional connections (see left panel of Figure 4). The structure of the graph was further examined in order to determine whether the method identified plausible network modules. Modularity-based clustering [1] was applied to the graph, resulting in 60 modules that exceeded the threshold size of 100 vertices. A number of neurobiologically plausible resting-state networks were identified, including "default mode" and sensorimotor networks (right panel of Figure 4). In addition, the method identified a number of structured coherent noise sources (i.e. MRI artifacts) in the dataset. For both neurally plausible and artifactual modules, the modules detected by BIGQUIC are similar to those identified using independent components analysis on the same dataset, without the need for the extensive dimensionality reduction (without statistical guarantees) inherent in such techniques.

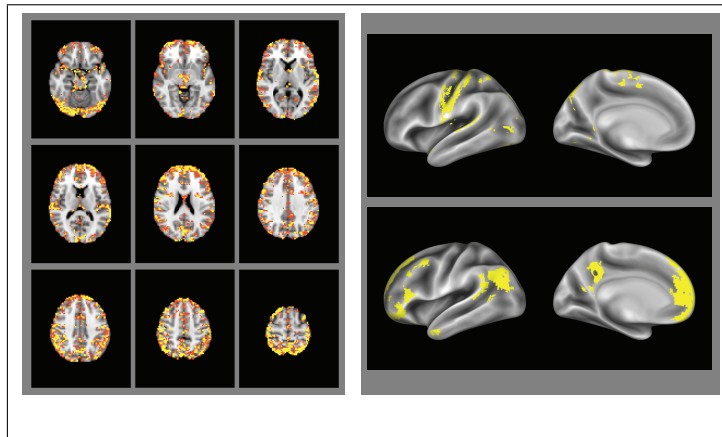

Figure 4: (Best viewed in color) Results from BIGQUIC analyses of resting-state fMRI data. Left panel: Map of degree distribution across voxels, thresholded at degree=20. Regions showing high degree were generally found in the gray matter (as expected for truly connected functional regions), with very few high-degree voxels found in the white matter. Right panel: Left-hemisphere surface renderings of two network modules obtained through graph clustering. Top panel shows a sensorimotor network, bottom panel shows medial prefrontal, posterior cingulate, and lateral temporoparietal regions characteristic of the "default mode" generally observed during the resting state. Both of these are commonly observed in analyses of resting state fMRI data.

## Acknowledgments

This research was supported by NSF grant CCF-1320746 and NSF grant CCF-1117055. C.-J.H. also acknowledges the support of IBM PhD fellowship. P.R. acknowledges the support of ARO via W911NF-12-1-0390 and NSF via IIS-1149803, DMS-1264033. R.P. acknowledges the support of ONR via N000140710116 and the James S. McDonnell Foundation.

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
