[Supplementary Material]

# 6 Appendix

The following elementary lemma shows that the residual controls the error of the CG method.

**Lemma 2.** *If $\hat{\mathbf{w}}_i$ satisfies $\|X\hat{\mathbf{w}}_i - \mathbf{e}_i\|_2 < \epsilon$, then $\frac{\epsilon}{\sigma_{max}(X)} < \|\hat{\mathbf{w}}_i - \mathbf{w}_i\|_2 < \frac{\epsilon}{\sigma_{min}(X)}$.*

*Proof.* We define the residual $b = X\hat{\mathbf{w}}_i - \mathbf{e}_i$, and note that the optimal solution $\mathbf{w}_i$ satisfies $X\mathbf{w}_i = \mathbf{e}_i$. Therefore, $b = X\hat{\mathbf{w}}_i - X\mathbf{w}_i = X(\hat{\mathbf{w}}_i - \mathbf{w}_i)$, which finishes the proof. □

## 6.1 Proof of Theorem 1

### 6.1.1 Background

We begin with a formal definition of the approximate Newton direction based on an approximate Hessian.

**Definition 1.** *Let $J$ denote a (symmetric) subset of variables. We define the Newton direction restricted to $J$ with a positive definite approximated Hessian $H$ as follows:*

$$D_J^H(X) \equiv \arg \min_{\substack{D:D_{ij}=0 \\ \forall (i,j) \notin J}} \operatorname{tr}(\nabla g(X)^T D) + \frac{1}{2} \operatorname{vec}(D)^T H \operatorname{vec}(D) + \lambda \|X + D\|_1. \qquad (10)$$

Since $H$ is positive definite, (10) is well-defined. For convenience, we also introduce $D_J(X) \equiv D_J^{\nabla^2 g(X)}(X)$ to denote the computation using the exact Hessian.

We work within the framework of [9], but in a more general setting encompassing approximate Newton directions. At each iteration, the iterate $Y_t$ is updated by $Y_t \leftarrow Y_t + \alpha_t D_{J_t}^{H_t}(Y_t)$, where $J_t$ is a subset of variables chosen at iteration $t$, and $H_t$ is the Hessian approximation. Under the setting of QUIC and BIGQUIC, $J_1, J_3, \dots$ denote the fixed sets, and $J_2, J_4, \dots$ denote the free sets. The sequence $J_t$ satisfies the following Gauss-Seidel condition:

$$\bigcup_{j=0,\dots,T-1} J_{t+j} \supseteq \mathcal{N} \quad \forall t = 1, 2, \dots \qquad (11)$$

where $\mathcal{N}$ is the set including all possible $(i, j)$ pairs. Also, we require

$$H_t \succ \eta \, \forall t = 1, 2, \dots, \qquad (12)$$

which is part of the assumption of Theorem 1. The setting is summarized in Algorithm 2.

---

**Algorithm 2:** General Block Quadratic Approximation method for Sparse Inverse Covariance Learning

---

**Input** : Empirical covariance matrix $S$ (positive semi-definite $p \times p$), regularization parameter matrix $\lambda$, initial $Y_0$, inner stopping tolerance $\epsilon$
**Output**: Sequence of $Y_t$.
1 **for** $t = 0, 1, \dots$ **do**
2     Generate a variable subset $J_t$.
3     Compute the Newton direction $D_t \equiv D_{J_t}^{H_t}(Y_t)$ by (10).
4     Compute the step-size $\alpha_t$ using an *Armijo*-rule based step-size selection in (6).
5     Update $Y_{t+1} = Y_t + \alpha_t D_t$.

---

### 6.1.2 Global convergence

We prove that the sequence $\{Y_t\}_{t=1,2,\dots}$ converges to the global optimum, thereby generalizing Theorem 1.

Proposition 3 in [9] states that the line search condition will be satisfied in a finite number of iterations, meaning that the step size $\alpha$ is well defined. Next, we generalize Lemma 2 and Proposition 2 of [9] to accommodate the approximated Hessian.

**Lemma 3.** $\delta = \delta_J(X)$ *in the line search condition* (6) *satisfies*

$$\delta = \text{tr}((\nabla g(X))^T D^{H_t}) + \lambda \|X + D^{H_t}\|_1 - \lambda \|X\|_1 \leq -\eta \|D^{H_t}\|_F^2. \qquad (13)$$

*where* $D^{H_t} = D_J^{H_t}(X)$ *is the minimizer of the* $\ell_1$*-regularized quadratic approximation defined in* (10).

The proof is the same as in [9] after we replace $\nabla^2 g(X)$ by $H_t$. The following lemma shows that $D^H$ is an indicator of optimality.

**Lemma 4.** *For a positive definite H, X is the optimal solution of* $f(X)$ *if and only if* $D^H(X) = 0$.

*Proof.* If $D^H(X) \neq 0$, then Lemma 3 shows that $\delta(X) < 0$. According to Proposition 3 in [9] there is a line search step $\alpha > 0$ such that $f(X + \alpha D^H(X)) - f(X) < \sigma \alpha \delta(X)$, implying that $X$ cannot be the optimal solution.

Conversely, if $D^H(X) = 0$, we want to show that any direction $D$ and step size $\alpha > 0$ does not result in a descent. Since 0 is the optimal solution of (10), we have

$$\alpha \, \text{tr}(\nabla g(X)^T D) + \frac{1}{2} \alpha^2 \, \text{vec}(D)^T H \, \text{vec}(D) + \lambda \|X + \alpha D\|_1 \geq \lambda \|X\|_1.$$

So

$$\alpha \, \text{tr}(\nabla g(X)^T D) + \lambda \|X + \alpha D\|_1 - \lambda \|X\|_1 \geq \frac{1}{2} \alpha^2 \, \text{vec}(D)^T H \, \text{vec}(D).$$

Since $g(X + \alpha D) - g(X) = \alpha \, \text{tr}(\nabla g(X)^T D) + o(\alpha)$, we have

$$\lim_{\alpha \downarrow 0} \frac{f(X + \alpha D) - f(X)}{\alpha}$$
$$= \lim_{\alpha \downarrow 0} \frac{\alpha \, \text{tr}(\nabla g(X)^T D) + o(\alpha) + \lambda \|X + \alpha D\|_1 - \lambda \|X\|_1}{\alpha}$$
$$\geq \lim_{\alpha \downarrow 0} \frac{\alpha^2 \, \text{vec}(D)^T H \, \text{vec}(D) + o(\alpha)}{\alpha} = 0.$$

We proved that no direction $D$ is a descent direction, thus $X$ is the global optimum. $\square$

We can further generalize Lemma 4 when a subset $J$ of variables is used. We first define the minimum-norm subgradient for $f$.

**Definition 2.** *We define the minimum-norm subgradient* $\text{grad}_{ij}^S f(X)$ *as follows:*

$$\text{grad}_{ij}^S f(X) = \begin{cases} \nabla_{ij} g(X) + \lambda_{ij} & \text{if } X_{ij} > 0, \\ \nabla_{ij} g(X) - \lambda_{ij} & \text{if } X_{ij} < 0, \\ \text{sign}(\nabla_{ij} g(X)) \max(|\nabla_{ij} g(X)| - \lambda_{ij}, 0) & \text{if } X_{ij} = 0. \end{cases}$$

Lemma 4 in [9] shows that $\text{grad}_{ij}^S f(X) = 0$ for all $(i, j) \in J$ if and only if $X$ is optimal in a problem constrained to the subset $J$. Combine with Lemma 4 to prove the following:

**Lemma 5.** *For a positive matrix H and a subset of indexes J,* $D_J^H(X) = 0$ *if and only if* $\text{grad}_{ij}^S f(X) = 0 \ \forall (i, j) \in J$.

Next, we look at a convergent subsequence $Y_{s_t}$ just as in [9].

**Lemma 6.** *For any convergent subsequence* $Y_{s_t} \to Y^*$, *we have* $D_{s_t}^{\bar{H}} \equiv D_{J_{s_t}}^{\bar{H}}(Y_{s_t}) \to 0$ *for some* $\bar{H} \succ \eta$.

*Proof.* There exists an infinite index set $\mathcal{T} \subseteq \{s_1, s_2, \ldots\}$ and $\mu > 0$ such that $\|D_t^{H_t}\|_F > \mu$ for all $t \in \mathcal{T}$. We can assume $\alpha_{s_t} < 1$ for all $s_t$ without loss of generality. By selecting the subsequence $s_t$ appropriately we can also assume that $H_{s_t} \to \bar{H}$ for some $\bar{\eta} \succ \bar{H} \succ \eta$.

Using the same arguments as outlined in the proof of Lemma 6 in [9], we get

$$(1 - \sigma)\bar{\eta}^{-2}\mu \leq O(\alpha_t \| D_t^{H_t} \|_F), \tag{14}$$

Again, by Lemma 3, we have

$$-\alpha_t \delta_t \geq \alpha_t \eta \| D_t^{H_t} \|_F^2 \geq \eta \alpha_t \| D_t^{H_t} \|_F \mu.$$

Since $\{\alpha_t \delta_t\}_t \to 0$, it follows that $\{\alpha_t \| D_t^{H_t} \|_F\}_t \to 0$. Combining with $H_t \to \bar{H}$ we have $\{\alpha_t \| D_t^{\bar{H}} \|_F\}_t \to 0$. Taking limit as $t \in \mathcal{T}$ and $t \to \infty$, we have

$$(1 - \sigma)\eta\mu \leq 0,$$

a contradiction which finishes the proof. □

We have the tools to prove Theorem 1.

Assume a subsequence $\{Y_t\}_{\mathcal{T}}$ converges to $\bar{Y}$. Since the choice of the index set $J_t$ selected at each step is finite, we can further assume that $J_t = \bar{J}_0$ for all $t \in \mathcal{T}$, considering an appropriate subsequence of $\mathcal{T}$ if necessary. From Lemma 6, $D_{\bar{J}_0}^{\bar{H}_{\bar{J}_0}}(Y_t) \to 0$ for some positive definite matrix $\bar{H}_{\bar{J}}$. Therefore $D_{\bar{J}_0}^{\bar{H}_{\bar{J}_0}}(\bar{Y}) = 0$. This implies

$$\text{grad}_{ij}^S f(\bar{Y}) = 0 \ \ \forall(i,j) \in \bar{J}_1.$$

We can then apply Gauss-Seidel condition (11) to show

$$\text{grad}_{ij}^S f(\bar{Y}) = 0 \ \ \forall(i,j) \in \bar{J}_t$$

for all $t$, and thus $\bar{Y}$ is the optimal solution of (1).

## 6.2 Proof of Theorem 2

Next, we focus on proving the convergence rate of the sequence $\{X_t\}_{1,2,\dots}$ generated by BIGQUIC. First, we show that under assumption (8), the updates of BIGQUIC will be equivalent to unconstrained Newton updates.

**Lemma 7.** *There exists a $T > 0$ such that*

$$sign((X_t)_{ij}) = sign((X^*)_{ij}) \ \ \forall i,j, t \geq T,$$

*where $sign(a)$ can be $1, -1, 0$.*

*Proof.* First, consider the cast that $X_{ij}^* = 0$. The optimality condition of (1) implies that $|\nabla_{ij}g(X^*)| \leq \lambda$. Combined with Assumption (8), we have $|\nabla_{ij}g(X^*)| < \lambda$. Since $X_t \to X^*$ as proved in the previous section, $g(X_t) \to g(X^*)$, so there exists a $T_{ij} > 0$ such that

$$|\nabla_{ij}g(X_t)| < \lambda \ \ \forall t \geq T_{ij}.$$

Therefore, from the definition of *fixed* set, $(i,j)$ will be always in the *fixed* set when $t \geq T_{ij}$.

For other $X_{ij}^* \neq 0$, there exists a $T_{ij}$ such that $|X_{ij}^* - (X_t)_{ij}| < |X_{ij}^*|/2$ for all $t \geq T_{ij}$, which implies $\text{sign}((X_t)_{ij}) = \text{sign}(X_{ij}^*)$.

Combining two cases, we can take $T = \max_{i,j} T_{ij}$ and finish the proof. □

Lemma 7 suggests that we can divide the index set into the following two partitions:

$$F = \{(i,j) \mid X_{ij}^* \neq 0\},$$
$$Z = \{(i,j) \mid X_{ij}^* = 0\}., \tag{15}$$

where $Z$ is always in the *fixed* set.

According to Proposition 3 in [9], the step size equals to 1 when $X_t$ is close to $X^*$. Therefore, after a finite number of iterations, BIGQUIC solves the following unconstrained subproblem:

$$\arg\min_{X, X_{ij}=0 \forall(i,j)\in F} \{-\log \det X + \text{tr}(SX) + \lambda\|X\|_1\} \equiv \bar{f}(X), \tag{16}$$

Moreover, by Lemma 7, after a finite number of iterations $X_{ij}$ will never change its sign, that implies that the update rule for BIGQUIC is equivalent to solving the problem

$$\arg\min_{X, X_{ij}=0 \forall (i,j)\in F}\{-\log\det X + \text{tr}(SX) + \lambda \sum_{i,j}\text{sign}(X_{ij}^*)X_{ij}\} \equiv \bar{f}(X) \qquad (17)$$

with Newton's method with step size one. The objective function of (17) is smooth and therefore we can apply theorems derived for Newton's method on smooth unconstrained optimization.

**Newton methods on smooth unconstrained optimization**

We mainly use the results from [4] that considers the case of inexact Newton methods for unconstrained smooth function. The inexact Newton method is defined as follows. At each iteration, $\mathbf{x}_{t+1} \leftarrow \mathbf{x}_t + \mathbf{s}_t$, where $\mathbf{s}_t$ satisfies

$$\nabla^2 f(\mathbf{x}_t)\mathbf{s}_t + \nabla f(\mathbf{x}_t) = \mathbf{r}_t, \text{ and } \frac{\|\mathbf{r}_t\|}{\|\nabla f(\mathbf{x}_t)\|} \le \eta_k. \qquad (18)$$

Theorem 3.3 in [4] shows that the algorithm converges linearly if $\eta_k$ is upper bounded.

**Theorem 4.** *Assume that $\nabla^2 f(\mathbf{x})$ is Hölder continuous with exponent $p$ ($0 < p \le 1$) and the inexact Newton iterates $\{\mathbf{x}_t\}$ converge to $\mathbf{x}^*$. It follows that $\mathbf{x}_t$ converges to $\mathbf{x}^*$ with rate at least $1 + p$ if and only if $\|\mathbf{r}_t\| = O(\|\nabla f(\mathbf{x}_t)\|^{1+p})$.*

A function $f(\mathbf{x})$ is Hölder continuous if and only if

$$|f(\mathbf{x}) - f(\mathbf{y})| \le C\|\mathbf{x} - \mathbf{y}\|^p. \qquad (19)$$

In our problem, $\nabla^2 \bar{f}(X) = \nabla^2 g(X) = X^{-1} \otimes X^{-1}$ and according to [9] $X_t$ will be in a bounded set such that $MI \succ X_t \succ mI$. Therefore the objective function $\bar{f}(X)$ is Holders continuous. We can write $\|\mathbf{r}_t\| = O(\|\nabla f(\mathbf{x}_t)\|^{1+p})$ more formally as follows:

$$\exists C > 0 \text{ such that } \frac{\|\mathbf{r}_t\|}{\|\nabla f(\mathbf{x}_t)\|^{1+p}} \le C. \qquad (20)$$

The inexact Newton method is different from BIGQUIC because we exactly solve the problem, but with approximated Hessian. However, The following derivation connects the two.

Assume $H_t$ is the approximate Hessian in BIGQUIC, and $\mathbf{d}_t$ is the solution of (3). As have shown earlier, after finite number of iterations BIGQUIC updates are equivalent to solving (17) with Newton's method, which implies

$$\mathbf{d}_t = -H_t^{-1}\nabla\bar{f}(X).$$

Substitute into (18) and we have $\nabla^2 \bar{f}(\mathbf{x}_t)\mathbf{d_t} + \nabla\bar{f}(\mathbf{x}_t) = \mathbf{r}_t$, so

$$\mathbf{r}_t = \mathbf{d}_t + \nabla^2\bar{f}(\mathbf{x})^{-1}\nabla\bar{f}(\mathbf{x})$$
$$= (\nabla^2\bar{f}(\mathbf{x})^{-1} - H_t^{-1})\nabla\bar{f}(\mathbf{x}_t).$$

Therefore

$$\frac{\|\mathbf{r}_t\|}{\|\nabla\bar{f}(\mathbf{x}_t)^{1+p}\|} \le \frac{\|(\nabla^2\bar{f}(X))^{-1} - H_t^{-1}\|}{\|\nabla\bar{f}(\mathbf{x}_t)\|^p}.$$

Consider the case that $p > 1$. Define

$$\Delta = \nabla^2 g(\mathbf{x}_t) - H_t,$$

we have

$$\|\nabla^2\bar{f}(\mathbf{x}_t)^{-1} - H_t^{-1}\| = \|\nabla^2\bar{f}(\mathbf{x}_t)^{-1} - \nabla^2\bar{f}(\mathbf{x}_t)^{-1}(I + \nabla^2\bar{f}(\mathbf{x}_t)^{-1}\Delta)^{-1}\|$$
$$= \|\nabla^2\bar{f}(\mathbf{x}_t)^{-1}\Delta + o(\Delta)\|$$
$$\le \|\nabla^2\bar{f}(\mathbf{x}_t)^{-1}\|\|\Delta\| + o(\|\Delta\|).$$

Now for BIGQUIC, $\nabla^2\bar{f}(\mathbf{x}_t) = X_t^{-1} \otimes X_t^{-1}$, so we have

$$\|\nabla^2\bar{f}(X_t)^{-1} - H_t^{-1}\| \le \sigma_{max}(X_t)^2\|\Delta\| + o(\|\Delta\|).$$

Table 1: The time and memory requirement for sparse Cholesky factorization.

| $p$ (dimensionality) | $\|A\|_0$ | $\|L\|_0$ (memory usage) | time (sec) |
|---:|---:|---:|---:|
| 100 | 888 | 893 | 0.01 |
| 500 | 5,094 | 17,494 | 0.01 |
| 1000 | 9,992 | 57,547 | 0.02 |
| 5000 | 19,960 | 1,327,992 | 3.15 |
| 10000 | 99,948 | 5,388,053 | 30.00 |
| 50000 | 500,304 | 130,377,362 | 3245.00 |

Condition (20) becomes

$$\frac{\|\mathbf{r}_t\|}{\|\nabla \bar{f}(X_t)\|^{1+p}} \leq \frac{(\sigma_{max}(X_t))^2\|\Delta\| + o(\|\Delta\|)}{\|\nabla \bar{f}(X_t)\|^p}.$$

Therefore, as long as

$$\|H_t - \nabla^2 \bar{f}(X_t)\| = O(\|\nabla \bar{f}(X_t)\|^p), \qquad (21)$$

$\mathbf{r}_t$ satisfies (20), and therefore $X_t \to X^*$ for all the variables in $F$ (free set) with rate at least $1 + p$.
Also, all the variables in the fixed set will have $(X_t)_{ij} = 0 = X^*_{ij}$ as shown in Lemma 7, so $X_t \to X^*$ with rate at least $1 + p$, thus proving the theorem.

### 6.3 Proof of Theorem 3

From Lemma 2, $\|\hat{\mathbf{w}}_i - \mathbf{w}_i\| = O(\|\mathbf{b}_i\|)$, so the error of computed and exact $W$ can be bounded by $\|\hat{W} - W\| = O(\|B\|)$. To further bound $\|\hat{H} - \nabla^2 g(X)\|$, notice that $\hat{H} = \hat{W} \otimes \hat{W}$ and $\nabla^2 g(X) = X \otimes X$. Assume $\Delta = \hat{W} - W$, then

$$\begin{aligned}
\|\hat{H} - \nabla^2 g(X)\| &\leq \|(W + \Delta) \otimes (W + \Delta) - W \otimes W\| \\
&\leq \max_{\|D\|=1} \|(W + \Delta)D(W + \Delta) - WDW\| \\
&= \max_{\|D\|=1} \|2\Delta DW + \Delta D\Delta\| \\
&\leq 2\|\Delta\|\|W\| + \|\Delta\|^2 \\
&\leq O(\|\Delta\|) = O(\|B\|).
\end{aligned}$$

Combined with Theorem 2 we complete the proof.

### 6.4 Demonstration of the scalability of sparse Cholesky factorization

Recall that in the line search step, the computational bottleneck is the checking of positive definiteness and the computation of the determinant of a sparse matrix with dimension that can reach a million. In the following we motivate the conclusion that this problem cannot be expected to be solved using sparse Cholesky factorization for matrices with dimension larger than 100,000.

The Cholesky factorization $L$ ($A = LL^T$) of a sparse matrix $A$ is usually sparse, but not as sparse as the original matrix. The sparsity depends on the permutation of indices, but there is no theoretical guarantees. The time and space complexity for sparse Cholesky factorization is proportional to $\|L\|_0$ and in the following experiment we demonstrate that $\|L\|_0$ grows quickly in our synthetic data.

We generate random positive definite $p$ by $p$ matrices with $p = 100, 500, 1000, 5000, 10000, 50000$, and the number of nonzero elements $\|L\|_0$ is shown in Table 1. We can observe that the number of nonzeros in $L$ (computed by MATLAB sparse Cholesky factorization with symamd to find the permutation) grows nonlinearly with $\|A\|_0$ and or $p$ and therefore other (better) solutions have to be explored. That well motivates our Schur based approach.

### 6.5 Other Difficulties.

Many other difficulties arise when dealing with large sparse matrices in the sparse inverse covariance estimation problem. Randomized coordinate descent can converge much faster than cyclic coordinate descent when solving (3). This behavior is not completely understood, but empirical evidence

(a) Comparison of coordinate descent implementations utilizing different degrees of randomness.

(b) Comparison of the effect of different stopping conditions used for the conjugate gradient method for gradient computation.

Figure 5: Demonstration of two difficulties in scaling QUIC to ultra high dimensional data. Figure 5(a) shows that the degree of randomness in the coordinate descent solver is crucial for fast convergence. Figure 5(b) shows that the accuracy of the conjugate gradient method is important in BIGQUIC. Both figures run on ER dataset with $\lambda = 0.5$.

supports it, see Figure 5(a). In our proposed algorithm we process the variables in blocks as described in detail in Section 3.1; however this blocking scheme removes some degree of randomness in coordinate selection. Empirical evidence suggests that the problem can be solved by clustering, as shown in Figure 6.

Yet another problem to be tackled is determining the proper stopping tolerance for conjugate gradient descent (CG), the sparse linear solver we employ to compute columns of the matrix $W$. In Figure 5(b), we conduct another experiment that the CG stopping tolerance of gradient $W = X^{-1}$ is varied from $10^{-3}$ to $10^{-9}$, and the Hessian computation is set to be very accurate ($10^{-13}$). The results show that with a lower accurate gradient computation, the solver cannot converge to the optimal solution, so that the accuracy highly depends on the stopping tolerance of CG of gradient. In comparison, the Hessian computation can be inaccurate, as shown in in Section 4.

## 6.6 The benefit of graph clustering algorithm

In addition to the empirical number we showed in the main paper, we further provide exact count of boundary nodes for each off-diagonal blocks in Figure 6(b). As shown in Figure 6(b), total number of boundary nodes is 83 in a dataset with $p = 693$, means we only need to compute $693 + 83$ columns of $W$ in one sweep; while using a random partition requires $693 \times 5$ column computations.

As discussed in Figure 5(a), the convergence will be slow if we apply block coordinate descent to destroy the randomness. However, with graph clustering partition, the off-diagonal elements are minimized, so the variables are more decoupled into each block. In the extreme case, when there are no boundary points, all the off-diagonal blocks of $D$ and $W$ are 0, the Newton subproblem (3) can be decomposed into $k$ subproblems, each for one diagonal block. So block-coordinate descent can converge in one iteration if all the blocks are exactly minimized. Even if there are few off-diagonal elements, after one sweep over blocks the solution can be very close to the optimum. Figure 6(a) shows the results.

## 6.7 Time Analysis

In this section we present a detailed time complexity analysis. We assume $k$ is the number of blocks used in the coordinate descent step, $T$ is the average number of CG iterations and $T_{\text{inner}}$ is the number of coordinate sweeps in one Newton iteration, and $T_{\text{outer}}$ is number of coordinate descent sweeps within a single block. Finally, $s$ is the size of the free set, and $m$ is the number of nonzeros in $X_t$. In step 1, BIGQUIC computes the gradient in order to partition the variables into the *fixed* and *free* sets; this takes $O(mTp)$ time for computing $W$ and $O(p^2 d)$ time for computing $S = YY^T$. The graph clustering algorithm in the step 3 requires $O(s + kp)$ flops. The block coordinate descent method needs $O((p + |B|)T_{\text{outer}}mT)$ time for computing columns of $W$, where $|B|$ is the number

(a) Convergence of different coordinate descent strategies.

(b) Number of boundary nodes for each block identified by graph clustering.

Figure 6: A demonstration of identifying blocks by clustering. Figure 6(a) shows that the convergence of BIGQUIC is close to QUIC when blocks are identified by graph clustering algorithms. Figure 6(b) presents the number of boundary nodes (number of column evaluations of $W$) is very small for each off-diagonal block, when blocks are identified by graph clustering.

of boundary nodes, and $O(pm)$ for computing $U_{S_q}$, and $O(T_{\text{inner}}p/k)$ for the coordinate descent updates themselves. Finally, the line search steps cost $O(spTL)$, where $L$ is the number of the line search steps. We can see that the time needed to compute the columns of $W$, $O((p+|B|)mTT_{\text{outer}})$, dominates the time complexity, which underscores the importance of minimizing the number of boundary nodes $|B|$ via our clustering scheme. BIGQUIC is summarized in Algorithm 1.