[Reviews · NeurIPS 2013]

Submitted by Assigned_Reviewer_4

This paper describes an approach to derive the L1 regularized Gaussian maximum likelihood estimator for the sparse inverse covariance estimation problem. The focus of this paper was to scale the previous algorithm QUIC to solve problems involving million of variables. They describe three innovations brought about by this new approach: inexact Hessians, better computation of the logdet function, and carefully selecting the blocks updated in their block coordinate scheme via a smart clustering scheme. The numerical results test their new method against the previous QUIC algorithm, GLASSO and ALM, showing improved performance on a few select problems.

Clarity: this paper is well written, and expresses their ideas in a straight forward manner, though readers will struggle if they have not read the original QUIC paper. Furthermore, it would be enlightening to see the effect of the CG solves, T_{outer} and T_{inner} on the performance of BIGQUIC. How are you selecting these parameters?

Originality: the strategies presented in this paper seem novel, and the presence of the graph clustering algorithm is intriguing.

Significance: QUIC has shown to perform quite well for the sparse inverse covariance problem presented in this paper, relying mostly on the special structure exhibited by the problem formulation. As a result, any strategies that would increase the applicability of this approach could prove to be very useful for those interested in solving these types of problems.

Quality: this paper expresses their ideas very well, and presents smart ideas in scaling up QUIC to solve much larger problems then in the previous paper.

However, the global convergence proof seems to be inconclusive, with two glaring problems that need to be addressed:

1. The proof does not cover the effect of the fixed-free set selection at the beginning of each iteration. Rather, the proof, which borrows many principles from the proof of the original QUIC algorithm, proves the result for J being the set of all variables, which is not the case for this algorithm.

2. It is unclear from the theoretical analysis how T_{outer} and T_{inner} are affecting the proof of convergence or the numerical results. The proof from QUIC seemed to require the exact solution of the sub-problems at each iteration. However, it is not apparent as to why the two-loop coordinate descent procedure of Algorithm 1 would guarantee to exactly solve that inexact model (with an inexact Hessian), particularly in T_inner inner sweeps and T_outer outer sweeps. If you are indeed solving the inner/outer blocks inexactly, then please highlight its effect on the convergence proof and the numerical results.
Summary: I believe this paper is interesting, and provides some advancement over the original QUIC algorithm, but the authors must provide more precise arguments supporting the claim of global convergence, in light of the active set identification phase, and the effect of inexactly solving the inner sub-problems.

Submitted by Assigned_Reviewer_5

This paper is an extension of QUIC method proposed for sparse inverse covariance estimation problems a couple of years ago and published at NIPS. The new method is heavily based on QUIC, which employs active set selection strategy, second-order estimation of the smooth part of the objective, and a coordinate descent scheme to optimize the subproblems. The new methods uses clever and carefully executed schemes to exploit structure of the sparse inverse covariance estimation problems to scale each operation (specifically coordinate descent) and to avoid storage of the dense inverse matrix to enable solution of large-scale problems. The resulting method indeed scales well beyond any other method for sparse inverse covariance problems.

The key improvement is based on the idea of identifying graph clusters in the current working set to decompose the the matrix into blocks of columns (nodes) over which to perform repeated passes of coordinate descent. This minimizes the need for computations and updates of the rest of the matrix. An existing and presumably effective graph clustering algorithm is employed.

Other clever techniques are used, such as computing the blocks of the inverse of the parameter matrix, using a CG method, with variable degree of accuracy. For gradient computation the inverse is computed more accurately than for the Hessian estimations. The convergence theory is provided to show under which conditions the "inaccurate" method retains convergence properties. The theory is not very surprising, as it basically requires some second-order quality of the approximation, to obtain quadratic convergence, even if it is an exact Taylor expansion.

The computation results are quite impressive. They show scalability of the methods for very large problems and also in multi-core setting.

Some minor comments:

Line 209: "number" is missing a "b".
Line 268-269 "embarassingly" parallelized sounds somewhat awkward, although it is clear what is meant.

Line 270-278: The are step numbers of Algorithm 1 referenced here, but no numbers are given in Algorithm 1.
Line 289 - there is an extra "of"

Summary: This paper is an extension of QUIC method proposed for sparse inverse covariance estimation problems a couple of years ago and published at NIPS. The new methods uses clever and carefully executed schemes to further exploit structure of the problem to scale each operation to scale to very large problems. The resulting method is quite a substantial improvement over the state-of-the-art.

Submitted by Assigned_Reviewer_6

Summary:
This paper presents several extensions to the QUIC algorithm for sparse inverse covariance estimation, which together enable the method to scale to much larger domains that previously possible, with a maximum demonstrated size of a 1 million by 1 million sparse covariance matrix. The main innovations are: 1) a block coordinate descent approach that first clusters the inverse covariance matrix into near-diagonal blocks and then performs an l1 regularized Newton method on the largest size blocks that fit into memory; 2) an incremental approach to computing the log-det term for step size computation, and 3) a method that only computes certain Hessian terms approximately (this and the gradient computations both use conjugate gradient methods, but Hessian terms can use fewer iterations). The authors apply the method to two large synthetic data sets and a real fMRI data set, and show the method is able to learn these large models on a single multi-core machine.

Comments:
Overall, I think that this is a very good paper. Since this is really an algorithmic paper, there may be some concern that it represents an incremental advance over the QUIC and DC-QUIC papers themselves, but given that the orders of magnitude here are quite significant (20,000^2 -> 1,000,000^2), I think that the methods described are definitely worthy of publication on their own.

A few comments/questions:
- Can the authors elaborate on how they went about choosing the regularization parameter for domains this large? Normally this is fairly straightforward via cross-validation, but given that a single run here can take many hours, I'm not quite clear how they went about actually finding the choices of lambda mentioned in Section 5. Especially since methods like QUIC tend to have many more non-zero parameters during the optimization than at the final solution, this seems worth discussing.

- Similar to the above, would it be possible to show or highlight the number of non-zeros during the optimization phase, and the amount of time this ends up taking?

- Is the choice of lambda to achieve degree 20 for the fMRI data actually reasonable? Can we verify that this is a reasonable choice via held-out log-likelihood, or some similar metric? Given that this real data set was by far the most impressive demonstration in my mind, I'm curious if the proposed approach is learning a good model here, or just solving a hard optimization problem. Right now the only analysis is qualitative, which is nice but not ultimately that convincing especially given that they are ultimately just described as "similar" to those discovered by ICA even though that requires first reducing the dimension.

- Given that these methods are starting to become quite complex and difficult to independently implement, I hope the authors consider releasing a public version of this code.

- Did the authors consider using multifrontal LLT factorizations, etc? Given that they're already using PARMETIS, there are a number of similar "embarrassingly" parallelizable parallel Cholesky factorizations, and for the sparsity levels considered it doesn't seem completely implausible that they might be able to fit factorizations into memory (though of course not the inverse itself). This is a fairly minor point, since I'm sure the authors have put a lot of engineering into these efforts, but I'm genuinely curious if there are interesting tradeoffs involved here. In my experience, CG methods can be widely variable (in terms of the number of iterations needed), for getting an accurate inverse, even with preconditioning.
Summary: The authors propose modifications to the QUIC algorithm that let it scale to 1 million by 1 million covariance matrices. Given the substantial increase in size over past work, this is a very impressive piece of work.
Author Feedback

Author rebuttal: We thank the reviewers for their kind comments and suggestions.

1. Reviewer 1: "it would be enlightening to see the effect of the CG solves, T_{outer} and T_{inner} on the performance of BIGQUIC. How are you selecting these parameters?"

The stopping tolerance of CG is ||r^t||<\epsilon*||r^0||, where r^t is the residual at the t-th iteration.
Since gradient has to be computed accurately, we set \epsilon=1e-10.
For the Hessian computations, we follow Theorem 3 and set \epsilon=||grad^S(X_t)||.
According to the arxiv version of QUIC paper, T_{outer} should grow with number of iterations. We use exactly the same setting as QUIC (T_{outer}=1+iter/3).
We set T_{inner}=3 in all our experiments.

The following are the training time to achieve 1e-5 accrate solution on a 20000 fmri dataset using different T_{outer} and T_{inner}:

T_{inner}=1, T_{outer}= 1+iter, cputime = 2415s
T_{inner}=1, T_{outer}= 1+iter/3, cputime = 1941s
T_{inner}=1, T_{outer}= 1+iter/5, cputime = 2054s
T_{inner}=3, T_{outer}=1+iter, cputime = 2745s
T_{inner}=3, T_{outer}=1+iter/3, cputime = 1582s
T_{inner}=3, T_{outer}=1+iter/5, cputime = 1891s
T_{inner}=5, T_{outer}=1+iter, cputime = 2581s
T_{inner}=5, T_{outer}=1+iter/3, cputime = 1458s
T_{inner}=5, T_{outer}=1+iter/5, cputime = 1315s

So generally T_{outer} cannot be too large or too small.
If T_{outer} is too large, QUIC takes too much time to solve each quadratic approximation;
if T_{outer} is too small, the convergence is slow.
However, T_{inner} can be large because each inner iteration is cheap (only taking O(p/k) time).

2. Reviewer 1 "The proof does not cover the effect of the fixed-free set selection at the beginning of each iteration. Rather, the proof, which borrows many principles from the proof of the original QUIC algorithm, proves the result for J being the set of all variables, which is not the case for this algorithm."

We thank the reviewer for paying attention to our technical details. Our proof actually covers the effect of the fixed-free set selection in the following manner: we first prove the general block coordinate descent algorithm (Algorithm 2 in appendix) converges, and then use the argument that under the "fixed/free" set selection, each Newton iteration is equivalent to the following two steps:

(1) perform update on the fixed set, which will not change any variable (can be proved by our fixed set selection criterion), and then
(2) perform update on the free set.

Therefore our algorithm is just a special case of Algorithm 2, and will converge to the global optimum. Please see our detailed discussions in Appendix 6.2.

3. Reviewer 1: "It is unclear from the theoretical analysis how T_{outer} and T_{inner} are affecting the proof of convergence or the numerical results ... If you are indeed solving the inner/outer blocks inexactly, then please highlight its effect on the convergence proof and the numerical results."

Yes, in the theorems we assume each subproblem is solved exactly. We will highlight this assumption in the theorems.

4. Reviewer 3: "Can the authors elaborate on how they went about choosing the regularization parameter for domains this large?"

For synthetic data, the ground truth sparsity is known, so we start from a large lambda and gradually decrease it until the sparsity of *the final solution* is close to the ground truth sparsity. In the FMRI dataset we tested our algorithm with large lambda and then gradually decrease lambda until the solution has desired sparsity. Since our algorithm is fast for large lambda, this selection process is not too time consuming.

5. Reviewer 3: "would it be possible to show or highlight the number of non-zeros during the optimization phase, and the amount of time this ends up taking?"

We list number of nonzeros during the optimization phase for three testing cases with p=20000:

random graph:
step1: 214156 step2: 196811 step3: 186681 step4: 186681 step5: 186681

chain graph:
step1: 61558 step2: 59878 step3: 59876 step4: 59876 step5: 59876

fmri:
step1: 315462 step2: 206680 step3: 206680 step4: 206680 step5: 137794 step6: 137404 step7: 144178 step8:144064 step9:144070 step10:144062

So in our test cases empirically the number of nonzero do not grow too much in intermediate steps.

6. Reviewer 3: "Is the choice of lambda to achieve degree 20 for the fMRI data actually reasonable? Can we verify that this is a reasonable choice via held-out log-likelihood, or some similar metric?"

The choice of degree 20 was based on suggestion by an expert; but we agree that it would be interesting to analyze the network estimated by setting the regularization parameter lambda via cross-validation.

7. Reviewer 3: "I hope the authors consider releasing a public version of this code."

Yes, we will release the code soon.

8. Reviewer 3: "Did the authors consider using multifrontal LLT factorizations, etc?"

This is a great point. We tried the sparse LLT factorization before using CG, but we found in practice number of nonzeros in L will grow very quickly as p increase, even when average degree of the matrix is fixed. For example, when we generate a random p by p matrix X with average degree about 10, we found the number of nonzeros in L (computed by state-of-the-art sparse Cholesky factorization) grows quadratically with the dimension, so that sparse Cholesky factorization becomes infeasible when p>100,000, even when the average degree is only 10.

p nnz(X) nnz(L) time(sec)
100 888 893 0.01
500 5094 17494 0.01
1000 9992 57547 0.02
5000 19960 1327992 3.15
10000 99948 5388053 30.00
50000 500304 130377362 3245.00

Therefore, we use CG to compute the inverse and use another approach to compute logdet of a large sparse matrix.